

# Dataset construction method of cross-lingual summarization based on filtering and text augmentation

Hangyu Pan, Yaoyi Xi, Ling Wang, Yu Nan, Zhizhong Su and Rong Cao

State Key Laboratory of Mathematical Engineering and Advanced Computing, Zhengzhou, China

## ABSTRACT

Existing cross-lingual summarization (CLS) datasets consist of inconsistent sample quality and low scale. To address these problems, we propose a method that jointly supervises quality and scale to build CLS datasets. In terms of quality supervision, the method adopts a multi-strategy filtering algorithm to remove low-quality samples of monolingual summarization (MS) from the perspectives of character and semantics, thereby improving the quality of the MS dataset. In terms of scale supervision, the method adopts a text augmentation algorithm based on the pretrained model to increase the size of CLS datasets with quality assurance. This method was used to build an English-Chinese CLS dataset and evaluate it with a reasonable data quality evaluation framework. The evaluation results show that the dataset is of good quality and large size. These outcomes show that the proposed method may comprehensively improve quality and scale, thereby resulting in a high-quality and large-scale CLS dataset at a lower cost.

## INTRODUCTION

Cross-lingual summarization (CLS) converts *texts*[1] in one language into *summaries* in another language to enable people to quickly and efficiently obtain information from *texts* written in unfamiliar languages. CLS research has evolved from pipeline approaches (*Leuski et al., 2003*; *Siddharthan & McKeown, 2005*; *Orǎsan & Chiorean, 2008*; *Wan, Li & Xiao, 2010*; *Wan, 2011*; *Yao, Wan & Xiao, 2015*; *Zhang, Zhou & Zong, 2016*; *Ayana et al., 2018*; *Wan et al., 2019*; *Ouyang, Song & McKeown, 2019*) to end-to-end approaches (*Duan et al., 2019*; *Zhu et al., 2019*; *Xu et al., 2020*; *Cao, Liu & Wan, 2020*; *Takase & Okazaki, 2020*; *Ladhak et al., 2020*; *Dou, Kumar & Tsvetkov, 2020*; *Yin et al., 2020*; *Zhu et al., 2020*; *Bai, Gao & Huang, 2021*; *Bai, Gao & Huang, 2021*; *Wang et al., 2021*). The end-to-end approach has introduced deep learning models, such as the Transformer model (*Vaswani et al., 2017*). Extensive work has shown that the quality and scale of annotated data directly affect the performance of deep learning models. Therefore, both the quality and scale of the CLS dataset are extremely important.

Currently, researchers have constructed some CLS datasets using the collection method (*Ladhak et al., 2020*; *Nguyen & Daumé, 2019*; *Fatima & Strube, 2021*) and the

Corresponding author
Yaoyi Xi, WIM_GY@163.com

[1] We use "text" to refer to a carrier of information in general, alongside the categories such as image and speech, and "*text*" to refer specifically to the input in the sample pair (text-summary) of automatic text summarization, which means that "*summary*" represents the output in the sample pair.

transformation method (*Ayana et al., 2018*; *Duan et al., 2019*; *Zhu et al., 2019*). The most representative dataset is NCLS (*Zhu et al., 2019*). Datasets obtained by the collection method are of higher quality. However, they are also more expensive and thus, they are small in scale. The transformation method builds CLS datasets from the datasets of other tasks at a low cost and with a guaranteed scale. However, datasets obtained by the transformation method contain more low-quality samples, which seriously affects the performance of the CLS methods. There are two reasons for this phenomenon. First, there are errors in the source dataset. For example, Zh2EnSum, the subset of NCLS, which is derived from LCSTS (*Hu, Chen & Zhu, 2015*), contains many *summaries* that are too abstract because of the characteristics of the microblog, as shown in Table 1. Second, there are errors in the in the transformation system, such as translation errors. Thus, it is challenging in CLS research to build high quality, large-scale datasets for a reasonable cost.

To address the problems of existing datasets and their construction methods, we propose a CLS dataset construction method based on filtering and text augmentation that jointly supervises quality and scale. In terms of quality supervision, this method uses the multi-strategy filtering algorithm (MSF) which includes the strategies of irrelevant word statistics, keyword statistics, and semantics measure to remove low-quality samples of monolingual summarization (MS). In terms of scale supervision, the method uses the text augmentation algorithm based on a pretrained model (TAPT) to increase the size of CLS datasets.

The evaluation results show that MSF can easily and effectively improve the quality of MS datasets, and that TAPT can increase scale with assured quality. These results may be used to improve the performance of CLS systems and build CLS datasets. The CLS dataset constructed by our method is of extremely high quality and large scale, which indicates that our method can comprehensively improve the quality and increase the scale at a lower cost.

The main contributions of this article are as follows.

1. We propose MSF to improve the quality of MS datasets. This method removes low-quality MS samples from the perspective of character and semantics. It is the first to automatically check the degree to which the *summary* reflects the content of its original *text*, and to realize the content comparison between non-parallel texts. The semantics measure strategy in MSF implements the similarity measure for non-parallel texts, which can be widely applied.

2. We propose TAPT to increase the size of the text data with quality assurance. TAPT uses the self-attention mechanism, which is good at capturing the internal correlation of data or features, to select the words to be replaced. TAPT also uses MLM, which is an unsupervised pre-training task of the pretrained model, to realize contextual dynamic synonym replacement, greatly improving the effect of text augmentation. Experimental results shows that fine-tuning mBART (*Liu et al., 2020*) with TAPT can achieve +19.83 ROUGE-1, +15.4 ROUGE-2, and +17.4 ROUGE-L for English-Chinese CLS and +1.49 ROUGE-1, +0.31 ROUGE-2, and +4.99 ROUGE-L for Chinese-English CLS compared to the previous best performance (*Zhu et al., 2019*). TAPT can be used in conjunction with any supervised CLS method to further improve the performance of CLS systems.

**Table 1  Samples of the LCSTS dataset.**

| LCSTS | |
|---|---|
| Text | Reference summary |
| 近日国家能源局公布了《可再生能源发电并网驻点甘肃监管报告》，报告是在国家能源局对甘肃进行 3 个月可再生能源发电监管之后形成的。《报告》显示甘肃省可再生能源发电并网存在诸多问题。 | 能源局监管甘肃可再生能源全省弃风率超 20%。 |
| 一辆小轿车，一名女司机，竟造成 9 死 24 伤。日前，深圳市交警局对事故进行通报：从目前证据看，事故系司机超速行驶且操作不当导致。目前 24 名伤员已有 6 名治愈出院，其余正接受治疗，预计事故赔偿费或超一千万元。 | 深圳机场 9 死 24 伤续：司机全责赔偿或超千万。 |
| 中国有句古话"养儿防老"，而这三十年来所执行的强制计划生育政策使得"养儿防老"变为了不可能，绝大多数成员的养老问题除了依靠社会力量之外别无他路。养老不光是老人们所面临的问题，老无所依使得未老的社会成员也开始惶恐不安。 | 俞天任：老龄化问题不解决将亡族灭种。 |

**Notes.**

The underlined text denotes content that appears in both the text and the summary. The italicized text denotes content that appears in the summary but not in the text and is unrelated to the text. The bold text denotes content that appears in the summary but not in the text and reflects key information.

3. We propose a general and effective dataset construction method of CLS based on filtering and text augmentation. This method guarantees the quality of CLS dataset, meets the requirement of its scale, and can also be used to build more CLS datasets. This method was used to build a high-quality and large-scale English-Chinese CLS dataset (En2Zh_Sum) with 2,830,266 samples, which can be directly used for future research.

# RELATED WORKS

## CLS dataset

The collection method and the transformation method are the current CLS dataset construction methods. The overview of common CLS datasets is shown in Table 2. The collection method refers to obtaining texts from resource-rich platforms, such as the Internet, and organizing them into CLS datasets. This process is shown in Fig. 1. *Ladhak et al. (2020)* collected multilingual CLS datasets from WikiHow (https://wikihow.org). *Nguyen & Daumé (2019)* collected multilingual CLS from Global Voices (https://globalvoices.org). *Fatima & Strube (2021)* collected English-German CLS datasets from Spektrum der Wissenschaft (https://www.spektrum.de) and Wikipedia (https://wikipedia.org).

The transformation method refers to automatically generating CLS datasets from datasets of other tasks through a transformation system. The process is shown in Fig. 2. *Ayana et al. (2018)* built an English-Chinese CLS dataset by translating the *summaries* of Gigaword (*Napoles, Gormley & Durme, 2012*) and DUC (*Over, Dang & Harman, 2007*), while *Duan et al. (2019)* built a Chinese-English CLS dataset by translating the *texts* of Gigaword and DUC. *Zhu et al. (2019)* built English-Chinese and Chinese-English CLS datasets by translating *summaries* of the CNN/Daily Mail (*Hermann et al., 2015*), and LCSTS (*Hu, Chen & Zhu, 2015*), using a filtering strategy based on ROUGE (*Lin, 2004*).

**Table 2  An overview of CLS datasets.**

| Dataset | Method type | Mode | Scale | Open source |
|---|---|---|---|---|
| *Ladhak et al. (2020)* | Collection | Auto+Manual | 18k[*] | All |
| *Nguyen & Daumé (2019)* | Collection | Auto + Manual | gv-snippet: 1k[*]<br>gv-crowd: 0.2k[*] | All |
| *Fatima & Strube (2021)* | Collection | Auto + Manual | W-CLS: 51k<br>S-CLS: 48k | All |
| *Ayana et al. (2018)* | Transformation | Auto | 3.8M | Not |
| *Duan et al. (2019)* | Transformation | Auto | 3.8M | Some |
| *Zhu et al. (2019)* | Transformation | Auto | En2ZhSum: 371k<br>Zh2EnSum: 1.7M | All |

**Notes.**
An asterisk (*) denotes that the dataset contains many sub-datasets with cross-lingual directions. The average size of all sub-datasets is used to represent the size of this dataset.

**Table 3  An overview of text augmentation algorithms.**

| Algorithm | Object | Model | Method |
|---|---|---|---|
| *Wei & Zou (2019)* | Word | – | Synonym replacement, random insertion, random exchange, random deletion |
| *Kobayashi (2018)* | Word | Bidirectional Language Model | Synonym replacement |
| *Wu et al. (2019)* | Word | BERT | Synonym replacement |
| *Yu et al. (2018)* | Text | – | Back-translation |
| *Xie et al. (2019)* | Text | – | Back-translation |
| *Hou et al. (2018)* | Text | Seq2Seq Model | Generate new texts |
| *Anaby-Tavor et al. (2019)* | Text | GPT-2 | Generate new texts |

## Text augmentation

Data augmentation is a method for generating a large amount of data from a small amount of data using semantic invariance as a criterion (*Schwartz et al., 2018*). Common text augmentation algorithms can be categorized as word-level and text-level. The overview of related research is shown in Table 3.

In word-level augmentation, *Wei & Zou (2019)* proposed easy data augmentation (EDA), which includes four operations: synonym replacement, random insertion, random exchange, and random deletion. *Kobayashi (2018)* proposed a contextual text augmentation that uses a bidirectional language model for contextual dynamic synonym replacement. *Wu et al. (2019)* replaced the bidirectional language model of *Kobayashi (2018)* with BERT (*Devlin et al., 2018*).

In text-level augmentation, *Yu et al. (2018)* used back-translation (BT) (*Sennrich, Haddow & Birch, 2016a*; *Sennrich, Haddow & Birch, 2016b*) for text augmentation in reading comprehension tasks. *Xie et al. (2019)* proposed unsupervised data augmentation (UDA) for unsupervised text augmentation using BT. Some studies used the natural language generation (NLG) model for augmentation. *Hou et al. (2018)* proposed a data augmentation framework based on a sequence-to-sequence (Seq2Seq) model for the text

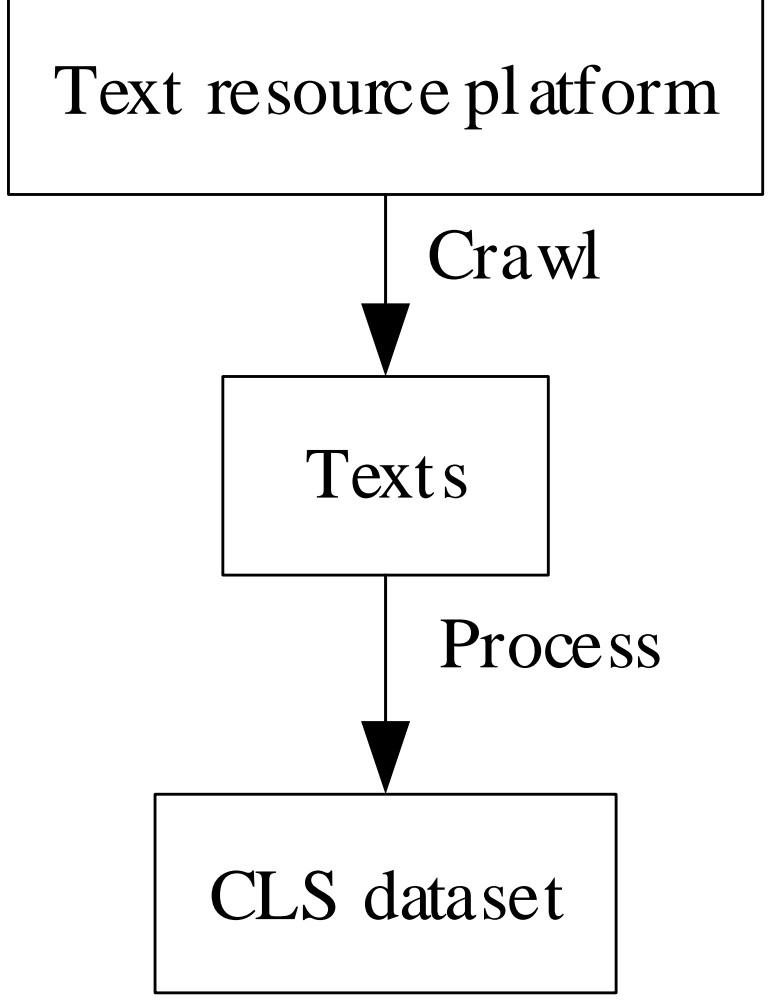

**Figure 1  The process of the collection method.**

augmentation of dialogue systems. *Anaby-Tavor et al. (2019)* proposed language-model-based data augmentation (LAMBDA), which used GPT-2 (*Radford et al., 2018*) to generate new texts for augmentation.

## METHODS

We propose a dataset construction method of CLS based on filtering and text augmentation to address the problems of existing datasets and their construction methods. This method applies MSF to improve the quality of the MS dataset, whose language is the target language of CLS (*text* in the source language, *summary* in the target language). Secondly, the method translates the *text* of the MS dataset into the source language and matches the translation with the corresponding *summary* of the original *text* to obtain a CLS dataset. Finally, the method uses TAPT to expand the sample pairs of the CLS dataset to obtain a high-quality

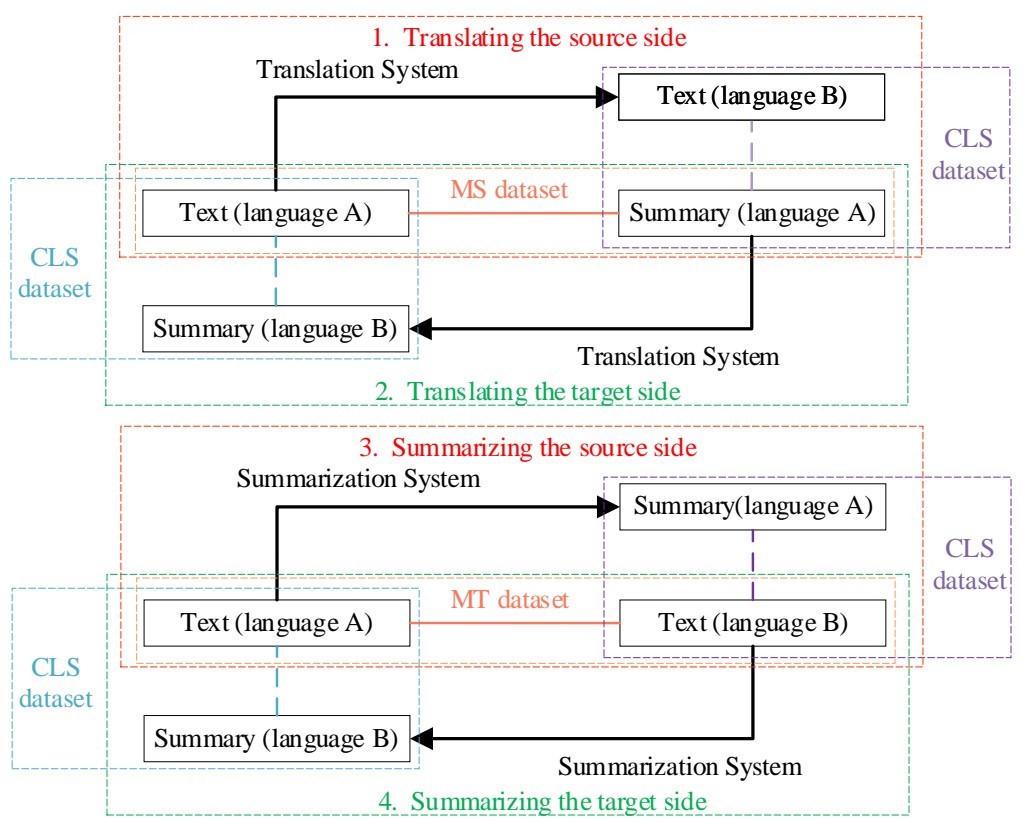

**Figure 2  The process of the transformation method.**

and large-scale CLS dataset. This method guarantees the quality of CLS dataset and meets the requirement of its scale. The process is shown in Fig. 3.

## Multi-strategy filtering

To accurately measure how well the *summary* in the MS dataset generalized the *text* content, we proposed a multi-strategy filtering algorithm. The algorithm improves the dataset quality successively using irrelevant word statistics, keyword statistics, and semantics measure strategies to remove low-quality MS sample pairs from the perspective of character, a combination of character and semantics, and semantics. The overall process is shown in Fig. 4.

## Irrelevant word statistics

The words in the *summary* that do not appear in its original *text* (defined as irrelevant words) will affect the learning effect of the CLS model to some extent. Therefore, this strategy calculates the proportion of irrelevant words in the *summary* to all *summary* words to measure how much *text* content the *summary* contains from the perspective of character. If the proportion is too high, it means that there are too many words in the *summary* that do not appear in the original *text*, and the sample should be filtered out.

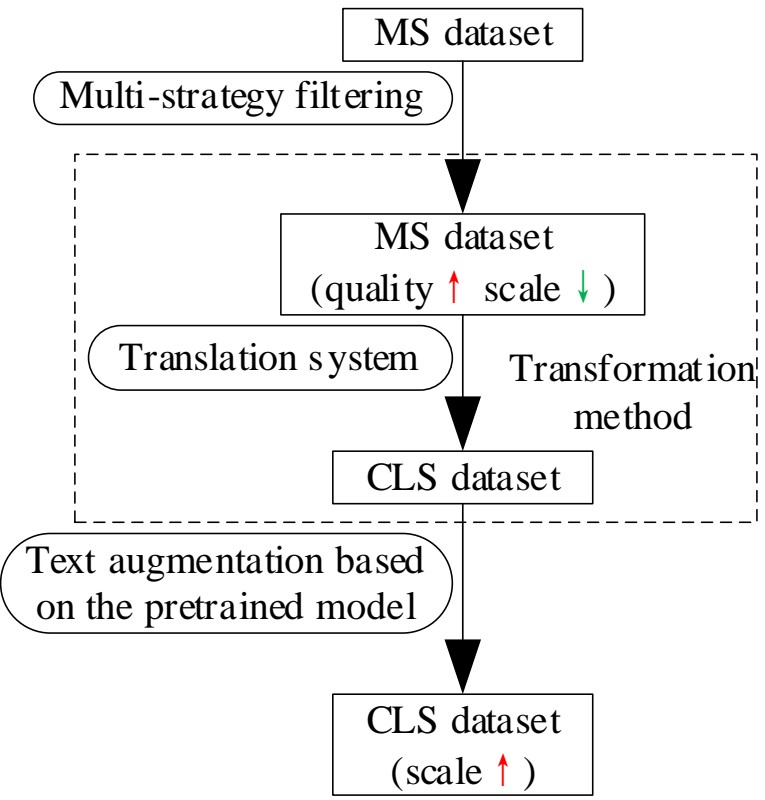

**Figure 3** **The process of the proposed dataset construction method of CLS.**

Specifically, given the *text* of an MS sample, $X = \{x_1, \ldots, x_i, \ldots, x_m\}$, and its reference *summary*, $Y = \{y_1, \ldots, y_j, \ldots, y_n\}$, $m$ is the length of $X$, $n$ is the length of $Y$, $n < m.x_i$ and $y_j$ denote the $i$th word of $X$ and the $j$th word of $Y$. Then, the proportion of irrelevant words $r_A$ is:

$$r_A = \frac{|\{y \in Y | y \notin X\}|}{n} \tag{1}$$

where $|\cdot|$ denotes the cardinal number of a set.

### Keyword statistics

A good *summary* should contain many keywords of the original *text*. Word embedding can reflect the semantic relationship of words in high-dimensional spaces and is a good choice for measuring semantic similarity to introduce semantic information (*Tang et al., 2019*). The K-means algorithm (*Macqueen, 1966*) can cluster similar objects into the same cluster. This strategy uses a word clustering method based on the Word2Vec (*Mikolov et al., 2013a*; *Mikolov et al., 2013b*) to extract keywords of a *text* from the perspective of semantics, and then calculates the proportion of words in a *summary* belonging to keywords of its corresponding *text* to all words in the *summary*. This will measure how much key information of the *text* is contained in the *summary* from the perspective of character. If

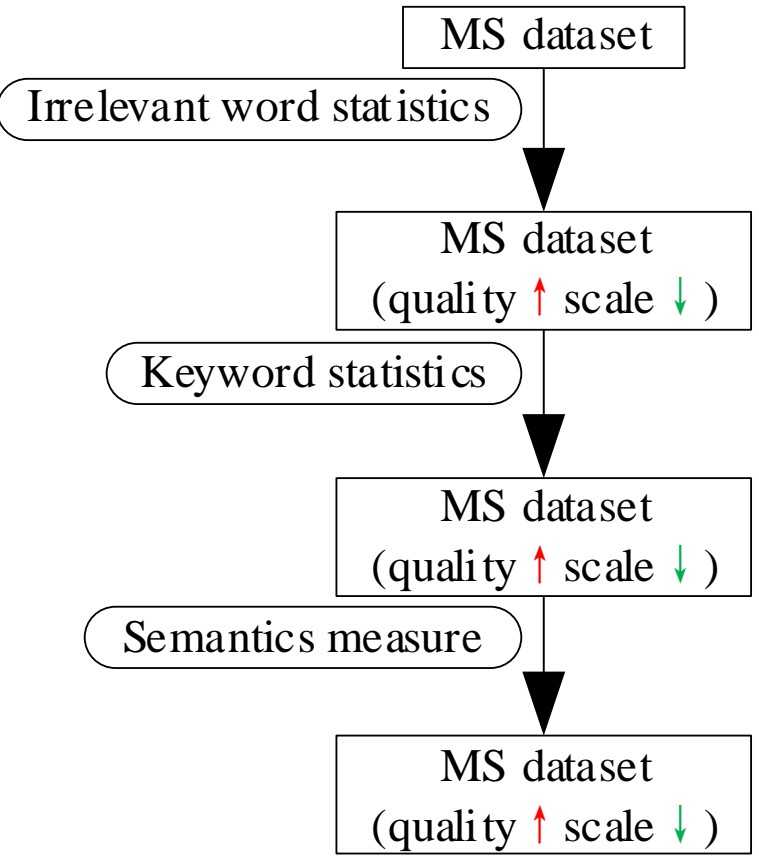

**Figure 4 The overall process of MSF.**

the proportion is too low, it means that the *summary* has too many non-keywords, and the sample should be filtered out.

Specifically, given $X$ and $Y$, we first encoded $X$ with Word2Vec to derive the word representation sequence $X = \{x_1, \ldots, x_i, \ldots, x_m\}$, and clustered all the words with the K-means algorithm. We then calculated the Euclidean distance between the cluster centers and other words, using the cluster centers as the main keywords, and selected the p nearest words to the cluster center as keywords to obtain the keyword set $C = \{c_1, \ldots, c_p\}$. The proportion of *summary* words belonging to keywords of the *text* $r_B$ is:

$$r_B = \frac{|\{y \in C\}|}{n} \tag{2}$$

where $|\cdot|$ denotes the cardinal number of a set.

## Semantics measure

A good *summary* should be semantically similar to the original *text*. Contextual word embeddings from the pretrained model, such as BERT (*Devlin et al., 2018*), have enhanced the semantic representation of texts. However, due to the problem of anisotropy, BERT-based text embedding cannot measure similarity using cosine similarity. BERT-whitening

**Table 4  Workflow of Whitening-h.**

**Algorithm 1 Whitening-h**

**Input:** Existing embeddings $\{z'_k\}^{2N}_{k=1}$ and reserved dimensionality h

1: compute Mean $\mu$ and variance $\Sigma$ of $\{z'_k\}^{2N}_{k=1}$

2: compute $U$, $\Lambda$ and $U^T = SVD(\Sigma)$

3: compute $W = (U\sqrt{\Lambda^{-1}})[:,:h]$

4: **for** $k = 1, 2, \ldots, 2N$ **do**

5: $\tilde{z}'_k = (z'_k - \mu)W$

6: **end for**

**Output:** Transformed embeddings $\{\tilde{z}'_k\}^{2N}_{k=1}$

(*Su et al., 2021*) solves the problem by transforming the embedding vector into isotropic form by whitening (*i.e.*, using principal component analysis). Therefore, this strategy takes BERT-whitening as text embedding, and calculates the cosine similarity between the representation vectors of the *text* and its *summary* to measure how much *text* content the *summary* contains from the perspective of semantics. If the cosine similarity is too small, the similarity between the *summary* and the *text* is too low, and the sample should be filtered out.

Specifically, given $X$ and $Y$, we first obtained the word representation sequences of $X$ and $Y$ by BERT word embedding, $X = \{x_1, \ldots, x_i, \ldots, x_m\}$ and $Y = \{y_1, \ldots, y_j, \ldots, y_n\}$, respectively. Their text representation vectors $x'$ and $y'$ were then obtained. The values $x'$ and $y'$ were unified and denoted as $z'$. $\{z'_k\}^{2N}_{k=1}$ was whitened and h principal components were retained to obtain $\{\tilde{z}'_k\}^{2N}_{k=1}$. The process is shown in Table 4 (*Su et al., 2021*). Finally, $\{\tilde{z}'_k\}^{2N}_{k=1}$ was split into $(\tilde{x}'_s, \tilde{y}'_s)^N_{s=1}$, and the cosine similarity $r_C$ between $x'$ and $y'$ was:

$$r_c = \cos(\tilde{x}', \tilde{y}') \tag{3}$$

where $\cos(\cdot)$ computes the cosine similarity of two vectors.

## Text augmentation based on the pretrained model

Self-attention (*Vaswani et al., 2017*) can capture inter-word dependencies. MLM, a pre-training task of auto-encoded pre-trained models such as BERT and RoBERTa (*Liu et al., 2019*), can contextually predict words. Therefore, we propose a text augmentation algorithm based on the pretrained model that uses the self-attention and MLM to dynamically replace synonym words for generating a new *text*.

Specifically, given the *text* of a CLS sample $X^{src} = \{x^{src}_1, \ldots, x^{src}_i, \ldots, x^{src}_m\}$ and its reference *summary* $Y^{tgt} = \{y^{tgt}_1, \ldots, y^{tgt}_j, \ldots, y^{tgt}_n\}$, we first used self-attention to select the words to be masked, obtaining $X^{src}_{masked} = \{x^{src}_1, \ldots, < mask >, \ldots, x^{src}_m\}$. Subsequently, we predicted the masked words using the MLM of the pretrained model to obtain the new *text* $X^{src'} = \{x^{src}_1, \ldots, x^{src'}_i, \ldots, x^{src}_m\}$. Finally, $X^{src'}$ and $Y^{tgt}$ were constructed together as a new CLS sample. The process is shown in Fig. 5, where blue text indicates that the predicted result is different from the original *text*, and green text indicates that the predicted result is the same as the original *text*.

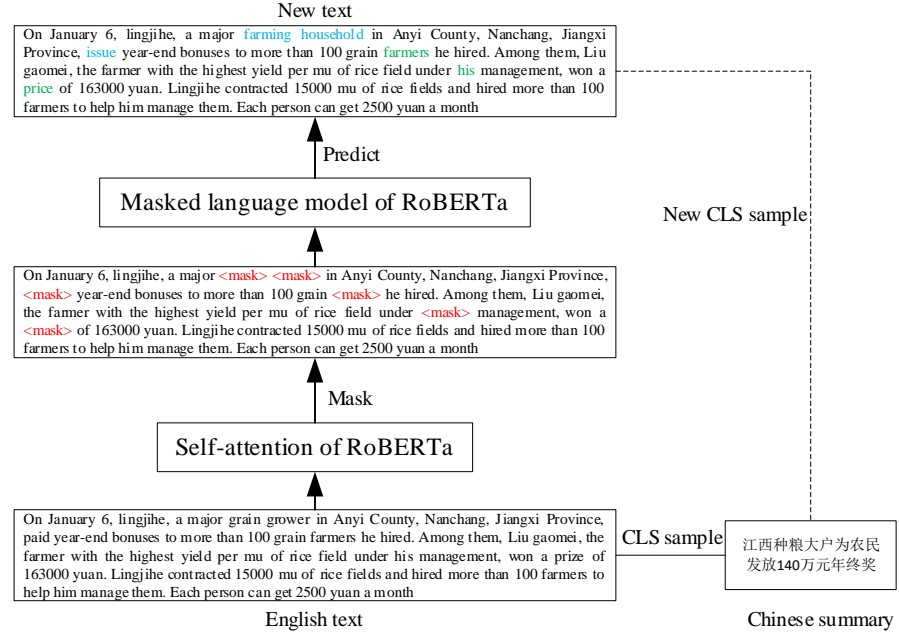

**Figure 5** The process of TAPT.

# EXPERIMENTAL SETUP

## Dataset

LCSTS (*Hu, Chen & Zhu, 2015*) is a Chinese summarization dataset originating from Sina Weibo, containing Part_I, Part_II, and Part_III. The authors scored samples of Part_II and Part_III to judge the relevance of the *summary* to the *text*. The correlation score interval is [1,5], and the higher the score, the more relevant it is. In this study, 2,196,263 samples from Part_I after deduplication and 195 samplesform Part_III with a score of 5 after deduplication were used as the original samples for building En2Zh_Sum.

NCLS (*Zhu et al., 2019*) is the benchmark set of CLS. We used it to validate TAPT. It contains the English-Chinese CLS dataset En2ZhSum and Chinese-English CLS dataset Zh2EnSum. The statistics are shown in Table 5; the word segmentation algorithm is BPE (*Sennrich, Haddow & Birch, 2016a*; *Sennrich, Haddow & Birch, 2016b*). LCSTS is the data source of Zh2EnSum. We randomly sampled one-sixth of the En2ZhSum training set (60,781 samples) and one-half of the Zh2EnSum training set (846,857 samples) due to the large data size, and considering the hardware, training effect, training efficiency, and other factors. TAPT was used to obtain the augmented training subsets, with the data size reaching 115,589 and 1,424,296 samples, respectively.

## Baselines and comparison methods

To validate TAPT, it was used directly with CLS and compared with other research results. However, the study of neural CLS is still nascent and there are not many research results at present. Some representative research results are as follows:

**Table 5  Statistics on the NCLS dataset.**

| En2ZhSum | Train | Valid | Test | Zh2EnSum | Train | Valid | Test |
|---|---|---|---|---|---|---|---|
| Num[a] | 364,687 | 3,000 | 3,000 | Num[a] | 1,693,713 | 3,000 | 3,000 |
| SrcAvgToken[b] | 942.7 | 949.1 | 930.2 | SrcAvgToken[b] | 73.4 | 73.3 | 73.6 |
| SrcMaxToken[c] | 12,498 | 7,547 | 8,635 | SrcMaxToken[c] | 134 | 113 | 119 |
| TgtAvgToken[d] | 70.0 | 70.1 | 69.9 | TgtAvgToken[d] | 20.6 | 20.6 | 21.5 |
| TgtMaxToken[e] | 593 | 242 | 260 | TgtMaxToken[e] | 70 | 48 | 53 |

**Notes.**
[a] Num denotes the size of the dataset.
[b] SrcAvgToken denotes the average token number of source language texts.
[c] SrcMaxToken denotes the maximal token number of source language texts.
[d] TgtAvgToken denotes the average token number of target language summaries.
[e] TgtMaxToken denotes the maximal token number of target language summaries.

*Zhu et al. (2019)* provided a benchmark for CLS studies and covers pipeline methods and end-to-end methods; it is described below.

**TETran**: Translates *texts* in the source language using a transformer-based MT model and then summarizes the translated *texts* in the target language using the LexRank algorithm (*Erkan & Radev, 2004*).

**TLTran**: Summarizes *texts* in the source language using a transformer-based MS model and then translates *summaries* in the source language to the target language using a transformer-based MT model.

**GETran** and **GLTran**: Replaces the MT model in TETran and TLTran with Google Translator (https://translate/google.com).

**NCLS**: Trains a Transformer (*Vaswani et al., 2017*) on NCLS.

**NCLS-MT**: Trains a Transformer by incorporating MT and CLS under multi-task learning.

**NCLS-MS**: Trains a Transformer by incorporating MS and CLS under multi-task learning.

The following summarizes other recent outstanding CLS studies.

**XNLG-CLS** (*Xu et al., 2020*): Fine-tunes the XNLG model (*Chi et al., 2020*) on NCLS.

**ATS** (*Zhu et al., 2020*): Trains a Transformer on NCLS, then summarizes the neural network probability distribution of the Transformer and the translation probability distribution of a probabilistic bilingual dictionary as the final *summary* generation distribution.

**MLPT** (*Xu et al., 2020*): Pretrains the CLS model using two unsupervised pretraining tasks and three supervised pretraining tasks, then fine-tunes the model by incorporating MS and CLS under multi-task learning.

**RL-XSIM** (*Dou, Kumar & Tsvetkov, 2020*): Uses a Transformer to perform multi-task learning for CLS, MT, and MS, and then optimizes the model through bilingual semantic similarity.

**MCLAS** (*Bai, Gao & Huang, 2021*): Modifies the output of CLS into sequential connections between MS and CLS.

**CSC** (*Bai et al., 2021*): Uses the compression ratio to unify the MT and CLS corpora, and encodes the compression ratio into the semantic representation of *texts*.

The above are the most representative research results of CLS at present. They were used them as the baselines for our study. The pretrained model BART (*Lewis et al., 2020*) had achieved state-of-the-art performance on MS at the time. Therefore, we chose the multilingual pretrained model mBART (*Liu et al., 2020*) as the basic framework of CLS, taking advantage of its powerful semantic understanding, cross-lingual alignment, and text generation capabilities. Combining the methods in this study, the following three comparison models were obtained.

**mBART-CLS**: Uses mBART directly for CLS.

**mBART$_{ft}$-CLS**: Fine-tunes mBARTon the train subsets of NCLS.

**(mBART+TPTA)$_{ft}$-CLS**: Fine-tunes mBARTon the augmented train subsets of NCLS.

## Parameter setup and evaluation metric
## Parameter setup

We used the transformation method to construct our dataset. We avoided introducing errors to the reference *summaries* that may have affected the learning effect of the CLS model by translating *texts* of LCSTS instead of *summaries*. We also used the Baidu Translate API (https://api.fanyi.baidu.com) as the transformation system to ensure the translation quality when building En2Zh_Sum. In MSF, we used the jieba library (https://pypi.org/project/jieba) for Chinese word segmentation, while the Word2Vec-based word clustering method was obtained from the Word2Vector in the gensim library (https://pypi.org/project/gensim) and the K-means algorithm in the sklearn library (https://pypi.org/project/sklearn). BERT embedding and whitening were performed using bert-base-uncased (https://huggingface.co/bert-base-uncased/tree/main) from Huggingface-transformers and codes from the NLP-Series-sentence-embeddings project (https://github.com/zhoujx4/NLP-Series-sentence-embeddings). The average word vector of all words in the first and last layers of the BERT word vector was used for text embedding. *Li et al. (2020)* proved that this pooling was the optimal choice without any processing. In TAPT, we used BPE (*Sennrich, Haddow & Birch, 2016a*; *Sennrich, Haddow & Birch, 2016b*) to tokenize[2] the texts and build a word dictionary. All of the English texts were used in lower case. Roberta-base (https://huggingface.co/roberta-base/tree/main) and mbart-large-cc25 (https://huggingface.co/facebook/mbart-large-cc25) from the Huggingface-transformers were used to implement RoBERTa and mBART.

The input/output sequence lengths were set to 550/100 and 80/60 for English-Chinese and Chinese-English CLS, respectively, to verify En2Zh_Sum and TAPT. The AdamW (*Loshchilov & Hutter, 2019*) optimizer was used to train in parallel on two NVIDIA RTX A6000 GPUs. Fine-tuning was stopped after 100,000 iterations. The key parameters of the experiments are shown in Table 6.

We also tested the performance of five classical pretrained models for predicting words to select the most appropriate pretrained model for TAPT, including BERT, ELECTRA (*Clark et al., 2020*), ERNIE (*Sun et al., 2020*), RoBERTA, and ALBERT (*Lan et al., 2020*). Specifically, the electra-base-discriminator (https://huggingface.co/google/electra-base-discriminator), ernie-2.0-base-en (https://huggingface.co/PaddlePaddle/ernie-m-large), and albert-base-v2 (https://huggingface.co/albert-base-v2/tree/main) models from the

[2] It will obtain tokens, which is the basic unit in which a computer processes text.

**Table 6  Key parameters of experiments.**

| Parameter | Setup |
|---|---|
| CLS Tokenizer[a] | BPE |
| En2Zh I/O length[b] | 550/100 |
| Zh2En I/O length[c] | 80/60 |
| Iter[d] | 100,000 |

**Notes.**
[a]Tokenizer denotes the tokenize algorithm.
[b]En2Zh I/O length denotes the input/output sequence length of model in English-to-Chinese CLS.
[c]Zh2En I/O length denotes input/output sequence length of the model in Chinese-to-English CLS.
[d]Iter denotes the iterations at the end of fine-tuning.

Huggingface-transformers were used to implement the pretrained model ELECTRA, ERNIE, and ALBERT, respectively.

## Evaluation metric

Artificial intelligence applications require large quantities of training and test data, which presents significant challenges concerning the availability of such data and its quality. Incomplete, erroneous, or inappropriate training data can lead to unreliable models that ultimately produce poor decisions (*Budach et al., 2022*). Therefore, a comprehensive and rigorous data quality assessment is important for dataset construction. The three quality attributes used for assessment are comprehensiveness, correctness, and variety, which are most critical to the "fit for purpose" of deep learning (*Chen, Chen & Ding, 2021*). We used qualitative or quantitative methods to evaluate the quality of the datasets produced by our dataset construction method using the three quality attributes. The data quality assessment framework proposed by *Chen, Chen & Ding (2021)* was used to qualitatively evaluate the comprehensiveness of the dataset by checking the data source, qualitatively evaluate the correctness of the dataset by manually checking samples, and quantitatively evaluate the variety of the dataset by checking the uniqueness of samples and the overlap of the train, validation, and test sets. According an example from *Chen, Pieptea & Ding (2022)*, we designed a group of experiments directly for CLS to quantitatively evaluate the effect of TAPT and the quality of data obtained by it.

We used ROUGE (*Lin, 2004*) to evaluate CLS results to verify En2Zh_Sum and TAPT, specifically, using the rouge-metric library (https://pypi.org/project/rouge-metric). The standard ROUGE metric only evaluates English *summaries* therefore, a special treatment was applied to evaluate Chinese *summaries* in our study, *i.e.,* the *summaries* were segmented by character granularity and then spliced with space characters.

In order to select the most appropriate pretrained model for TAPT, we used the average accuracy of predicted words equal to the masked words to measure the predictive power of pretrained models.

**Table 7  Human evaluation results on the three datasets.**

| Dataset | Role | Split | CR[a] | CC[a] | FL[a] |
|---------|------|-------|-------|-------|-------|
| LCSTS | Source | Train | 3.48 | 3.80 | 4.08 |
| | | Valid | 3.56 | 3.79 | 4.01 |
| | | Test | 3.62 | 3.83 | 4.03 |
| LCSTS$_{MSF}$ | Intermediate | Train | 4.10 | 3.77 | 4.05 |
| | | Valid | 4.05 | 3.84 | 4.09 |
| | | Test | 4.09 | 3.81 | 4.02 |
| En2Zh_Sum | Final | Train | 4.08 | 3.78 | 4.12 |
| | | Valid | 4.12 | 3.86 | 4.04 |
| | | Test | 4.06 | 3.82 | 4.02 |

**Notes.**
[a]CR, CC, and FL denote the scores for correlation, conciseness, and fluency, respectively.
LCSTS$_{MSF}$ represents the samples left after MSF is used on the LCSTS dataset.

# EXPERIMENTAL RESULTS AND ANALYSIS

## Evaluation of dataset quality
### Check of the comprehensiveness

In order to check the comprehensiveness of the data, it is important to evaluate the data collection procedure and data sources (*Chen, Chen & Ding, 2021*). The process of our dataset construction method is shown in Fig. 3. Firstly, we used MSF to remove the low-quality samples from the data source, ensuring quality at the beginning of the construction. Then, we used the Baidu Translation service to translate the *text* in the data source from Chinese to English, ensuring the quality of the collection procedure. Finally, we used TAPT to expand the CLS dataset obtained in the previous step, which increases the data size while ensuring the sample quality. We selected the LCSTS (*Hu, Chen & Zhu, 2015*) dataset as the data source. LCSTS is a benchmark dataset of ATS obtained from Sina Weibo. Its texts are short and noisy, which not only makes the model easier to learn from, but also increases the generalization performance. *Hu, Chen & Zhu (2015)* manually marked the correlation between the *text* and the *summary*. This correlation reflects the quality of samples. We can select samples with different correlation scores according to specific tasks, so as to obtain the validation set and test set of appropriate quality. The above qualitative assessment is sufficient to prove that En2Zh_Sum is of good comprehensiveness and reliable quality.

### Check of the correctness

The most straightforward way to check the correctness of a dataset is to check the sample data manually (*Chen, Chen & Ding, 2021*). Therefore, we randomly tested 100 samples from the train, validation, and test set of En2Zh_Sum and checked them manually. Three graduate students were asked to check each sample from three independent perspectives: (1) correlation, (2) conciseness, and (3) fluency. Each perspective was assessed with a score ranging from 1 (worst) to 5 (best). Table 7 presents the average results.

As shown in Table 7, the *summaries* and their corresponding *texts* had good conciseness and fluency. In the LCSTS$_{MSF}$ and En2Zh_Sum, *summaries* reflected the content of their

**Table 8  Checking results of the uniqueness and overlap of En2Zh_Sum splits.**

| Split | Uniqueness ratio | Overlap ratio |
|-------|------------------|---------------|
| Train | 100% | 0% (with Valid) |
| Valid | 100% | 0% (with Test) |
| Test | 100% | 0% (with Train) |

corresponding *texts*. However, in LCSTS, the correlation between the *summaries* and their corresponding *texts* was obviously low. The En2Zh_Sum had good correctness and reliable quality. The increased correlation score from LCSTS to LCSTS$_{MSF}$ indicates the effect of MSF on improving the quality of MS data set.

## Checking the variety

The unique data items in a dataset and the overlap in the train, validation, and test sets are the properties of the variety that must be checked (*Chen, Chen & Ding, 2021*). We calculated the uniqueness ratio of the train, validation, and test sets of En2Zh_Sum, as well as their overlap ratio. Table 8 presents the checking results, which show that the samples in En2Zh_Sum are unique, and there is no overlap among the three splits. Therefore, En2Zh_Sum is of good variety and reliable quality.

## Experimental evaluation

The experimental study in machine learning and deep learning can quantitatively evaluate the quality of the dataset (*Chen, Pieptea & Ding, 2022*). We fine-tuned mBART on the augmented train subsets of NCLS and compare the models of many CLS studies training on the full train set. The experimental results are listed in Table 9.

The experimental results show that the direct application of mBART does not perform well for either English-Chinese or Chinese-English CLS. These results suggest that the performance of a pretrained model cannot be directly applied to CLS without learning from specific data, even if that model is well-trained. mBART$_{ft}$-CLS (the mBART fine-tuned on the train subset) achieved +18.77 ROUGE-1, +13.2 ROUGE-2, and +15.84 ROUGE-L for English-Chinese CLS and +1.42 ROUGE-1, +0.11 ROUGE-2, and +4.98 ROUGE-L for Chinese-English CLS, compared to the state-of-the-art performance. These results show that the pretrained model can significantly improve the performance of the CLS system. (mBART+TPTA)$_{ft}$-CLS (the mBART fine-tuned on the augmented train subset) achieved +19.83 ROUGE-1, +15.4 ROUGE-2, and +17.4 ROUGE-L for English-Chinese CLS and +1.49 ROUGE-1, +0.31 ROUGE-2, and +4.99 ROUGE-L for Chinese-English CLS, compared to the state-of-the-art performance. These results indicate that TAPT can generate high-quality CLS samples, improve CLS performance, and indirectly validates the quality of En2Zh_Sum.

We can see that after fine-tuning the CLS task on the mBART, its performance is well above the baseline. It is difficult to improve the performance beyond this point. The essence of data augmentation to improve performance is to increase the samples in the train set. mBART$_{ft}$-CLS learned the train set well, while (mBART+TPTA)$_{ft}$-CLS was provided more training samples. Therefore, (mBART+TPTA)$_{ft}$-CLS should not have a significant

**Table 9  The results of CLS experiments.**

| Method | English-to-Chinese CLS | | | Chinese-to-English CLS | | |
|---|---|---|---|---|---|---|
| | ROUGE-1 | ROUGE-2 | ROUGE-L | ROUGE-1 | ROUGE-2 | ROUGE-L |
| -Pipeline methods- | | | | | | |
| TETran | 26.15 | 10.60 | 23.24 | 23.09 | 7.33 | 18.74 |
| TLTran | 30.22 | 12.20 | 27.04 | 33.92 | 15.81 | 29.86 |
| GETran | 28.19 | 11.40 | 25.77 | 24.34 | 9.14 | 20.13 |
| GLTran | 32.17 | 13.85 | 29.43 | 35.45 | 16.86 | 31.28 |
| -End-to-end methods- | | | | | | |
| NCLS | 36.82 | 18.72 | 33.20 | 38.85 | 21.93 | 35.05 |
| NCLS-MT | 40.23 | 22.32 | 36.59 | 40.25 | 22.58 | 36.21 |
| NCLS-MS | 38.25 | 20.20 | 4.76 | 40.34 | 22.65 | 36.39 |
| XNLG-CLS | 39.85 | 24.47 | 28.28 | 38.34 | 19.65 | 33.66 |
| ATS | 40.47 | 22.21 | 36.89 | 40.68 | 24.12 $^{\dagger}$ | 36.97 |
| MLPT | 43.50 $^{\dagger}$ | 25.41 $^{\dagger}$ | 29.66 | 41.62 $^{\dagger}$ | 23.35 | 37.26 $^{\dagger}$ |
| RL-XSIM | 42.83 | 23.30 | 39.29 $^{\dagger}$ | – | – | – |
| MCLAS | 42.27 | 24.60 | 30.09 | 35.65 | 16.97 | 31.14 |
| CSC | – | – | – | 40.30 | 21.43 | 35.46 |
| -The proposed method- | | | | | | |
| mBART-CLS | 14.59 | 4.31 | 10.87 | 0.71 | 0.04 | 0.70 |
| mBART$_{ft}$-CLS | 62.27$^{*}$ | 38.61$^{*}$ | 55.13$^{*}$ | 43.04$^{*}$ | 24.23$^{*}$ | 42.24$^{*}$ |
| (mBART + TAPT)$_{ft}$-CLS | **63.33** | **40.81** | **56.69** | **43.11** | **24.43** | **42.25** |

**Notes.**

ROUGE F1 scores (%) on En2ZhSum and Zh2EnSum test sets. A cross (†) denotes the previous best performance. An asterisk (*) denotes the results of fine-tuning MBART on the train subsets. The bold number denotes the results of fine-tuning MBART on the augmented train subsets.

performance improvement over mBART$_{ft}$-CLS. However, the results unexpectedly showed that the performance improved approximately 1% and 0.1% for English-Chinese and Chinese-English datasets, respectively. The bi-direction performance has a big difference. There are two main reasons: (1) mBART is a multilingual pretrained model. Due to the differences in the pre-training corpus and the characteristics of Chinese and English, the language ability of the model was different. This model can be regarded as two different models when conducting CLS experiments in two different cross-lingual directions. (2) The datasets for bidirectional CLS experiments were distinct. The dataset used for English-Chinese CLS was En2ZhSum, and the dataset used for Chinese-English CLS was Zh2EnSum. The statistics are shown in Table 5. Their source, size, length of samples, and other aspects have clear disparities. Therefore, it is quite normal for two different pretrained models to have distinct results for different datasets.

The size of En2Zh_Sum is shown in Table 10. To simply and intuitively validate the quality of En2Zh_Sum, we randomly sampled one-seventh of the train set (400,000 samples) to fine-tune mBART and conduct testing on the whole test set. The results are shown in Table 11. The results indicate that the CLS model can perform well with only a portion of En2Zh_Sum, which proves the quality of our dataset, En2Zh_Sum, and the effectiveness and feasibility of the dataset construction method of CLS.

**Table 10  Data size of the En2Zh_Sum.**

| En2Zh_Sum | Train | Valid | Test |
|---|---|---|---|
| Size | 2,810,266 | 10,000 | 10,000 |

**Table 11  ROUGE F1 scores (%) on the En2Zh_Sum test set.**

| Model | English-Chinese CLS | | |
|---|---|---|---|
|  | ROUGE-1 | ROUGE-2 | ROUGE-L |
| mBART$_{ft}$-CLS | 46.30 | 23.80 | 42.45 |

### Choice of the pretrained model

We randomly sampled five English texts from NCLS, and randomly selected ten words from each text, as shown in Table 12. We then used five pre-trained models (BERT, ELECTRA, ERNIE, RoBERTa, and ALBERT) to predict the masked tokens. The average prediction accuracy is shown in Table 13.

The experimental results show that RoBERTa had the highest accuracy, which indicates that it had the optimal performance for predicting words. Table 14 shows two samples of the results of applying RoBERTa in TAPT. The result of the first text is the same as the original text, and the result of the second text is slightly different from the original text. Therefore, RoBERTa can ensure both similarities and differences between the generated text and the original text to generate suitable new samples for augmentation.

One confusing result is that ERNIE's performance was 0. Table 13 shows the average accuracy of predicted words equal to the masked words to measure the predictive power of the model. The average accuracy is the mean of the ratio of the number of predicted words equal to the masked words to the total number of masked words in all experimental samples. ERNIE did not get a single word right, so the average accuracy was 0. ERNIE is a very powerful pretrained model, which improves the MLM of BERT and although the performance of ERNIE on various NLP tasks is greatly improved, the experimental result shows that its ability to predict words directly actually decreased, which is unsuitable for TAPT.

## CONCLUSIONS

We proposed a dataset construction method of CLS that jointly supervises its quality and scale, and we built a high-quality, large-scale English-Chinese CLS dataset called En2Zh_Sum. Our method used MSF to remove low-quality MS samples from the perspectives of character and semantics to supervise quality, and TAPT, which uses self-attention and MLM to increase samples to supervise scale. The experimental results showed that our method can comprehensively filter out low-quality samples and augment data scale, flexibly and effectively, to obtain a high-quality and large-scale CLS dataset at a lower cost.

Currently, there are few methods to evaluate and improve the quality of MS datasets. MSF is the first method to improve the quality of MS datasets by measuring the degree to

**Table 12  The experimental data.**

| Text | Masked token |
|------|-------------|
| According to [MASK] latest Reuters news, the U.S. police updated the number of casualties in the Denver shooting [MASK] 12 deaths and 58 injuries. On Friday night local time, 30 [MASK] people were [MASK] hospitalized for treatment, [MASK] of whom were in [MASK] condition. [MASK] 24-year-old [MASK] James Egan Holmes is being interrogated and [MASK] motive has not [MASK] determined yet. Compiled and reported by CNTV Jiang Yiyi. | 'the', 'as', 'injured', 'still', '11', 'critical', 'The', 'suspect', 'his', 'been' |
| Robin Lee, member of [MASK] CPPCC National Committee [MASK] CEO [MASK] Baidu, [MASK] that his proposal this [MASK] mainly [MASK] on using the Internet to improve the current network registration system. He [MASK] that the restrictions on commercial institutions to [MASK] out online registration business in some [MASK] should be lifted, and the allocation of medical [MASK] should be optimized with the help of social forces | 'the', 'and', 'of', 'revealed', 'year', 'focused', 'suggested', 'carry', 'regions', 'resources' |
| According [MASK] the news on the 21st, the continuous rainstorm caused [MASK] torrents at k806 + 500 of national highway [MASK] in Guangyuan, Sichuan, and some roads were damaged. At present, it is impossible to predict the opening time. At about 6:00 on the 21st, flash floods [MASK] out at Tashan Bay on national highway 212, [MASK] about [MASK] meters of asphalt concrete subgrade was washed away, [MASK] local uplift [MASK] the pavement and subsidence of the [MASK] Edited and [MASK] by CCTV yanghanning. | 'to', 'mountain', '212', 'broke', 'and', '600', 'with', 'of', 'subgrade.', 'reported' |
| From now on, the Municipal Bureau of urban and rural planning [MASK] launched [MASK] overall conceptual planning solicitation activity [MASK] 15 xiangjiangzhou islands. The overall conceptual planning solicitation of xiangjiangzhou Island [MASK] two [MASK] at the same time, [MASK] the International Solicitation [MASK] world-class professional design units and the solicitation for [MASK] schemes" for the public. For details, please visit the official website of the Municipal Bureau of [MASK] and rural [MASK] | 'has', 'an', 'for', 'opened', "channels", 'namely', 'for', "good", 'urban', 'planning.' |
| Liang [MASK] a lawyer from Zhonglun law [MASK] suggested that female [MASK] should [MASK] the police at the first time. As for the [MASK] of applying glue to long hair, which is [MASK] infringement [MASK] physical rights in civil law, although it is bad, it has not risen to the level of crime in [MASK] It can only be imposed with administrative penalties [MASK] as fines and criticism and education in accordance with [MASK] law on public security administration and punishment. | 'Jing', 'firm', 'victims', 'call', 'act', 'an', 'of', 'law.', 'such', 'the' |

**Notes.**

[MASK] indicates that the token at this position is masked.

**Table 13  The average accuracy of predictions.**

| Model | Accuracy |
|-------|----------|
| BERT | 0.44 |
| ELECTRA | 0.42 |
| ERNIE | 0 |
| RoBERTA | 0.5 |
| ALBERT | 0.24 |

which the *summary* reflects the content of its original *text* from the perspectives of character and semantics. It is simple and effective, and can be generalized to handle similar types of non-parallel text pairs. Compared with existing text augmentation algorithms based on pretrained models, TAPT utilizes self-attention to more rationally select words to be replaced. In the dynamic synonym replacement, TAPT uses a more powerful pre-training model to get the best performance of predictive words. TAPT encourages researchers to make reasonable use of the features of pretrained models, and can be used to augment texts for other tasks. Our dataset construction method is the first systematic method to build

**Table 14   Results of the RoBERTa-based TAPT.**

| Original text | Generated text |
|---|---|
| By the end of last year, the balance of broad money (M2) in China had reached 97.42 trillion yuan, and there was no doubt that it would exceed one billion yuan . This figure is 1.5 times that of the United States, 4.9 times that of Britain and 1.7 times that of Japan. This figure is close to a quarter of the total global money supply. It is no exaggeration to say that China has become the largest country in the global money stock | By the end of last year, the *balance* of broad money (M2) in China had reached 97.42 trillion yuan, and there was no doubt *that* it would exceed one billion *yuan* . This *figure* is 1.5 *times* that of the United States, 4.9 times that of Britain and 1.7 times that of Japan. This figure is close to a quarter of the total global *money* supply. *It* is no *exaggeration* to say that China has become the *largest* country in the global money *stock* |
| It was learned from authoritative sources yesterday that Zhong\'an online property insurance company, jointly established by Alibaba\'s Jack Ma, Ping An\'s Jack Ma and Tencent\'s Jack Ma, has now completed the regulatory approval process. It is expected that the CIRC will officially issue an approval document approving its preparation soon. It is reported that Eurasia Ping, a mysterious rich businessman , will take the post of chairman, which is jointly recommended by the ''three horses'' | It was learned from authoritative *sources* yesterday that Zhong\'an online *property insurance* company, jointly *established* by Alibaba\'s Jack Ma, Ping An\'s Jack Ma and Tencent\'s Jack Ma, has now *completed* the regulatory *approval* process. It is expected that the CIRC will officially issue an **official** document approving its *preparation* soon. It is *reported* that Eurasia Ping, a mysterious rich *businessman* , will take the role of chairman, which is jointly recommended by the ''three horses'' |

**Notes.**
Underlined words denote the masked words. Italicized words denote the same prediction result as the original words. Bold words denote a different prediction result from the original words.

CLS datasets, which adopted effective techniques to strictly supervise the quality and scale, and can be directly used to build more CLS datasets for future research.

In future studies, we will optimize our method's supervision process for quality and scale. In terms of quality supervision, we intend to more accurately measure how well the *summary* reflects the content of the original *text* from the perspective of semantics. In terms of scale supervision, we will consider how best to leverage the capabilities of the pretrained model to expand our samples with higher quality.

# ACKNOWLEDGEMENTS

We thank the reviewers for their helpful comments, PeerJ for English copyediting services and editage for linguistic assistance during the preparation of this manuscript.

### Funding
This work was supported by the National Social Science Foundation of China (No. 19CXW027). There was no additional external funding received for this study. The funders had no role in study design, data collection and analysis, decision to publish, or preparation of the manuscript.

### Grant Disclosures
The following grant information was disclosed by the authors:
National Social Science Foundation of China: No. 19CXW027.

### Competing Interests
The authors declare there are no competing interests.

### Author Contributions

- Hangyu Pan conceived and designed the experiments, performed the experiments, analyzed the data, performed the computation work, prepared figures and/or tables, authored or reviewed drafts of the article, and approved the final draft.
- Yaoyi Xi conceived and designed the experiments, analyzed the data, authored or reviewed drafts of the article, and approved the final draft.
- Ling Wang performed the experiments, performed the computation work, prepared figures and/or tables, and approved the final draft.
- Yu Nan performed the experiments, authored or reviewed drafts of the article, and approved the final draft.
- Zhizhong Su analyzed the data, prepared figures and/or tables, and approved the final draft.
- Rong Cao performed the computation work, authored or reviewed drafts of the article, and approved the final draft.

### Data Availability
The LCSTS: A Large-Scale Chinese Short Text Summarization Dataset is available at http://icrc.hitsz.edu.cn/Article/show/139.html. Researchers must request access to the dataset by sending an application to the dataset owners: Baotian Hu: baotianchina@gmail.com, Qingcai Chen: qingcai.chen@hit.edu.cn.

The Datasets for Cross-Lingual Summarization (NCLS) are available at Github and Zenodo: https://github.com/ZNLP/NCLS-Corpora; Hangyu Pan, Yaoyi Xi, Ling Wang, Yu Nan, Zhizhong Su, & Rong Cao. (2023). Dataset construction method of cross-lingual summarization based on filtering and text augmentation [Data set]. Zenodo. https://doi.org/10.5281/zenodo.7694044.

The code is available in the Supplemental Files.

## Supplemental Information

Supplemental information for this article can be found online at http://dx.doi.org/10.7717/peerj-cs.1299#supplemental-information.

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
