# Peer review of "Dataset construction method of cross-lingual summarization based on filtering and text augmentation"

_PeerJ Computer Science, doi:10.7717/peerj-cs.1299_

## Round 0.1 · original submission · Major Revisions

Dear Authors,

Please revise and resubmit your manuscript, particularly the English language. Also, kindly specify the key contributions of this work. Thank you.

·

Basic reporting

Dataset construction is critical but challenging for many NLP tasks. I believe s strict data quality evaluation strategy is important in the process. I would suggest the authors add some more references regarding data quality for machine learning datasets, for example,

(1) Chen, H., Chen, J., & Ding, J. (2021). Data evaluation and enhancement for quality improvement of machine learning. IEEE Transactions on Reliability, 70(2), 831-847.

(2) Chen, H., Pieptea, L. F., & Ding, J. (2022). Construction and Evaluation of a High-Quality Corpus for Legal Intelligence Using Semiautomated Approaches. IEEE Transactions on Reliability.

(3) Budach, L., Feuerpfeil, M., Ihde, N., Nathansen, A., Noack, N., Patzlaff, H., ... & Naumann, F. (2022). The Effects of Data Quality on Machine Learning Performance. arXiv preprint arXiv:2207.14529.

Experimental design

The experiment design is insufficient and more experiments need to be conducted to validate the quality of the dataset.

What is the contribution of this paper? The papers seem reused all the pertained language models without any modification.

Validity of the findings

The findings are doubtable: (1) In the results of CLS experiments, why the bi-direction performance has such a big difference? (2) In table 6, the ERNIE performance is 0? I doubt the experiment settings (3) In table 8, the testing set is too small compared to the training and validation sets.

Additional comments

I suggest the authors to submit the article to Frontiers in Big Data under the topic "Data Quality for Big Data and Machine Learning": https://www.frontiersin.org/research-topics/35131/data-quality-for-big-data-and-machine-learning

I feel the paper is more fit for this journal.

Reviewer 2 ·

Basic reporting

- Some minor reorganization of the paper is advised.

Experimental design

N/A

Validity of the findings

N/A

Additional comments

The author proposed an interesting and useful method for CLS dataset construction. The current version of the manuscript has some merits for publication. However, there are a few major issues that need to be addressed. I have the following suggestions for authors.
1. The abstract needs some improvement with facts and results.
2. The related work can be summarized in the form of a table.
3. The Dataset needs more elaboration. For example, why the authors choose 6:1 ratio for training in one dataset while 1:1 in another dataset?
4. The authors need to add some discussion on baseline models. For example, why do they consider a particular method as a baseline? Also, the authors need to clearly mention which methods are baseline and which are in comparison cohorts/proposed.
5. The performance improvement between MBART-CLS and MBART+PTA-CLS is minimal (around 1%). Is there any specific reason? Also, is this performance statistically significant? The authors need to perform some statistical analysis to further consolidate the results.
6. A detailed discussion of the results is required.
7. Is there any analysis of the effect of the filtering strategy for baseline as well as the proposed method? An ablation study may supplement this question.

Reviewer 3 ·

Basic reporting

1、Please carefully check English grammar and expression patterns to avoid inappropriate words. 2、Obviously, I hope to see more pictures in the article to show your work.

Experimental design

1.The authors should clearly elaborate on the differences between the proposed method and related works. 2.The method proposed in this paper already contains enough information to help other researchers. Of course, if possible, I also hope the author can disclose the details in GitHub. This is only a suggestion, not a necessity.

Validity of the findings

The experimental steps and methods are detailed, and the experimental results are also obvious. Of course, I would like all other formulas, pictures and tables to appear in the same document for better review.

---

## Round 0.2 · Minor Revisions

Dear Authors,

Please improve the English language presentation and quality of the figures in this manuscript. Thank you.

Reviewer 2 ·

Basic reporting

The manuscript needs english improvement. A proficient english speaker can help here.

Experimental design

N/A

Validity of the findings

N/A

Additional comments

The authors addressed all the concern raise in the previous round of review. The paper can be accepted for publicaiton after some english improvement.

Reviewer 3 ·

Basic reporting

The authors have addressed my concerns in the revised version. Therefore, I would like to give an accept to this paper.

Experimental design

no comment

Validity of the findings

no comment

Additional comments

The author should improve the quality of the pictures in the manuscript, and the font size should meet the specifications before the official publication.

---

## Round 0.3 · accepted · Accept

The comments have been addressed.